# A Scoping Review of the Association of Social Disadvantage and Cerebrovascular Disease Confirmed by Neuroimaging and Neuropathology

**DOI:** 10.3390/ijerph18137071

**Published:** 2021-07-02

**Authors:** Sarah A. Keller, Kellia J. Hansmann, W. Ryan Powell, Barbara B. Bendlin, Amy J. H. Kind

**Affiliations:** 1Department of Population Health Sciences, School of Medicine and Public Health, University of Wisconsin, Madison, WI 53705, USA; 2Center for Health Disparities Research, School of Medicine and Public Health, University of Wisconsin, Madison, WI 53705, USA; rpowell@medicine.wisc.edu (W.R.P.); bbb@medicine.wisc.edu (B.B.B.); ajk@medicine.wisc.edu (A.J.H.K.); 3Health Services and Care Research Program, Department of Medicine, School of Medicine and Public Health, University of Wisconsin, Madison, WI 53705, USA; 4Department of Family Medicine and Community Health, School of Medicine and Public Health, University of Wisconsin, Madison, WI 53705, USA; kellia.hansmann@fammed.wisc.edu; 5Department of Medicine, Geriatrics Division, School of Medicine and Public Health, University of Wisconsin, Madison, WI 53705, USA; 6Wisconsin Alzheimer’s Disease Research Center, School of Medicine and Public Health, University of Wisconsin, Madison, WI 53705, USA; 7Geriatric Research Education and Clinical Center (GRECC), William S. Middleton Hospital, United States Department of Veterans Affairs, Madison, WI 53705, USA

**Keywords:** cerebrovascular disease, social disadvantage, disparities, neuropathology, socioeconomic status, race, ethnicity, education

## Abstract

Social disadvantage—a state of low-income, limited education, poor living conditions, or limited social support—mediates chronic health conditions, including cerebrovascular disease. Social disadvantage is a key component in several health impact frameworks, providing explanations for how individual-level factors interact with interpersonal and environmental factors to create health disparities. Understanding the association between social disadvantage and vascular neuropathology, brain lesions identified by neuroimaging and autopsy, could provide insight into how one’s social context interacts with biological processes to produce disease. The goal of this scoping review was to evaluate the scientific literature on the relationship between social disadvantage and cerebrovascular disease, confirmed through assessment of vascular neuropathology. We reviewed 4049 titles and abstracts returned from our search and included records for full-text review that evaluated a measure of social disadvantage as an exposure variable and cerebrovascular disease, confirmed through assessment of vascular neuropathology, as an outcome measure. We extracted exposures and outcomes from 20 articles meeting the criteria after full-text review, and described the study findings and populations sampled. An improved understanding of the link between social factors and cerebrovascular disease will be an important step in moving the field closer to addressing the fundamental causes of disease and towards more equitable brain health.

## 1. Introduction

Social disadvantage—a state of low-income, limited education, poor living conditions, or limited social support—is associated with many chronic health conditions, including cerebrovascular disease. While age-adjusted death rates for cerebrovascular disease are declining in the overall population, significant socioeconomic disparities persist. Previous studies have identified associations between socioeconomic factors, including income [1], wealth [2], education [3,4,5], and occupation [6], and cerebrovascular disease and its risk factors including smoking and hypertension [7].

Despite an abundance of evidence underscoring these relationships, the mechanisms responsible for these associations remain unclear. The brain plays a central role in the processing of stress and the adverse conditions associated with social disadvantage [8,9]. In addition, the built environment and associated exposures may also be relevant to cerebrovascular outcomes, given the regional differences in stroke prevalence [10] and associations between air pollution exposure and stroke [11]. Defining the process that translates these social exposures into cerebrovascular disease is a necessary step in improving treatment and prevention, as well as eliminating disparities.

Definitive markers of cerebrovascular disease are most easily accessed through neuroimaging or autopsy. In addition to events such as ischemic or hemorrhagic stroke, which primarily involve macrovessels, cerebral small vessel disease may result in ischemic injury and also precede stroke [12]. Defining the causal mechanisms that lead to stroke requires a comprehensive assessment of cerebrovascular injury. Thus, the use of imaging techniques, including computed tomography (CT) and magnetic resonance imaging (MRI) scans or assessment via autopsy, provides tools for the comprehensive evaluation of cerebrovascular disease. Assessing the association between social disadvantage and cerebrovascular disease, confirmed through neuroimaging or neuropathological assessment, could provide important insight into how social context interacts with biological processes to produce disease.

As such, the objective of the current study was to evaluate the existing evidence regarding the relationship between life course social disadvantage and cerebrovascular disease, confirmed through imaging or autopsy. Due to the different ways in which social disadvantage can be defined, we chose to approach our evaluation of the literature through a scoping review. Whereas a systematic review is appropriate for summarizing the relevant information pertaining to a specific research question, a scoping review outlines the literature in a broader topic area [13], making it the best method to approach our objective.

To help frame our findings, we utilized the National Institute on Aging Health Disparities Research Framework [14], which was developed to assess the progress being made in aging research addressing health disparities and to identify areas requiring further inquiry. It is structured into four levels of analysis, which organize the many potential determinants of health disparities seen in older adults: environmental, sociocultural, behavioral, and biological. We used this framework to consider the different levels of context impacting health in each study we identified.

We summarized the current literature investigating potential associations between social disadvantage and confirmed cerebrovascular disease, including their data sources, methodologies, specific measures used, and their findings, as well as determining target areas for future research.

## 2. Materials and Methods

We used the systematic methodology of a scoping review to structure our search of the literature for records meeting our criteria. This allowed us to perform a rigorous literature search with transparency in our methods, without limiting our search to a single research question or particular study design in the returned records [15]. We followed the PRISMA extension for scoping reviews to complete this review and report our findings [16].

### 2.1. Identification of Relevant Studies

We systematically searched PubMed (Medline), PsychINFO, and the Sociology Research Database (SocINDEX) in collaboration with a trained research librarian. We limited our search to articles published in English, beginning in January 1990 through August 2020, in peer-reviewed journals. We created a comprehensive set of search terms related to cerebrovascular disease, social disadvantage, and health disparities by first identifying key terms through a precursory review of the literature, then using medical subject headings (MeSH) to expand on our cerebrovascular disease terms (Appendix A).

### 2.2. Study Selection

Figure 1 describes the article selection process. After the first author removed any duplicates, we identified records to review by the titles and abstracts of each reference. Our initial inclusion criteria required that article titles and abstracts explicitly mention an indicator or a composite of indicators of social disadvantage as an exposure variable. We defined social disadvantage as a state of low-income, limited education, poor living conditions, or limited social support. We also included studies that compared outcomes by race, to capture the ways systemic discriminatory practices and policies continue to cluster social disadvantage exposures among minoritized groups [17,18].

Our criteria at this stage also required that each record had a primary outcome of cerebrovascular disease, which we defined as any condition affecting blood flow and blood vessels in the brain, such as stroke, aneurysms, and stenosis. The first author queried 4049 returned records in more detail via abstract review to assess if they met the criteria that the cerebrovascular disease outcome was confirmed by brain imaging or autopsy. We excluded articles if they were not original research, such as a systematic review, if the study had children or animals as subjects, or if it was clear from the abstract that the cerebrovascular disease was not confirmed by brain imaging or autopsy. If it was not clear how the cerebrovascular disease was ascertained from the abstract, we included these records in the full text review.

### 2.3. Data Extraction and Reporting

After reviewing articles by title and abstract, the first and second authors independently reviewed the full text of 27 articles that met the criteria in the previous stage to confirm eligibility. All disagreements or uncertainties were resolved through a consensus meeting to determine the final list of included studies. The first and second authors extracted relevant characteristics during this process, including the study design, study population, and main findings. We examined these characteristics to identify patterns, themes, gaps, and areas to target in future research.

## 3. Results

Our search identified 4049 unique records, of which we found 27 eligible for full text review (Figure 1). Our criteria in the title and abstract review stage excluded 1315 records for having no social disadvantage exposure, 2217 records for having no cerebrovascular disease outcome confirmed by imaging or autopsy, 246 records for having an ineligible study population, such as children or animals, and 244 records which were not original research (e.g., review articles). Among the 27 records deemed appropriate for full text review, we ultimately excluded five articles for not clearly defining their measures of social disadvantage, and two articles for not confirming their cerebrovascular outcome measures through imaging or autopsy.

Our final review included 20 articles investigating the impact of social disadvantage on cerebrovascular disease outcomes confirmed by brain imaging or autopsy [19,20,21,22,23,24,25,26,27,28,29,30,31,32,33,34,35,36,37,38] (Table 1). The majority of the identified studies were published in the past decade (*n* = 11) [20,22,25,26,29,30,31,33,35,36,38], indicating growing interest in this kind of work. Nine of the included studies used a study sample based in the United States [20,21,23,24,27,32,34,36,38]; five were based in Europe, sampling from the United Kingdom [19,29,37], France [26], and Finland [31]; four were based in Asia, sampling from Japan [25], Taiwan [28], Singapore [35], and Siberia, Russia [22]; one used an Australian study population [30]; and one included subjects from both the US and Mexico [33]. More than half of the studies (*n* = 11) drew their data from a prospective cohort study or stroke registry [19,20,25,26,29,31,33,35,36,37,38], five used other cross-sectional observational designs [22,27,28,32,34], and the remaining used retrospective medical record reviews in their analysis [21,23,24,30]. The median sample size for the included studies was 722.

### 3.1. Cerebrovascular Outcomes

There was substantial variation among the cerebrovascular outcomes measured. Sixteen articles examined stroke, either generally (*n* = 12) [20,22,24,25,26,29,31,33,34,35,37,38] or focusing on hemorrhagic (*n* = 3) [21,27,30] or ischemic (*n* = 1) [19] subtypes. One article measured microbleeds as their primary outcome [23], two used indicators of vascular dementia [28,32], and one article measured white matter hyperintensities [36]. Fifteen articles used brain imaging (CT and/or MRI scans) alone to confirm the cerebrovascular outcomes [19,20,22,23,24,26,28,30,31,32,33,34,35,36,38]. Three studies confirmed the cerebrovascular disease with autopsy in addition to brain imaging, when appropriate [21,25,27], and the remaining two studies included cerebrospinal fluid analyses as well as autopsies and imaging [29,37].

### 3.2. Socioeconomic Disadvantage Assessments

We examined the specific measures of social disadvantage used by the studies in our sample, as well as the levels of analysis used by the investigators to frame their approach to studying the impact of social disadvantage on cerebrovascular outcomes, through the lens of the NIA Health Disparities Research Framework [14]. The distribution of the studies in our sample across the four levels of analysis provided by the framework are presented in Table 2. We discuss how these studies fit into this framework below.

#### 3.2.1. Environmental

A total of 11 studies in our sample evaluated their exposures as determinants occurring at an environmental level [19,20,26,28,29,30,31,32,34,36,37]. This level of analysis can include factors related to geography and politics, such as segregation and exposure to toxins; socioeconomic factors; income and occupation; and health care related factors, including health care access and literacy.

Area-based indices of disadvantage are an excellent example of exposures evaluated at this level, as they use a selection of measures to assess the level of disadvantage in a precise geographic area, rather than relying on individual-level measures, which may be difficult to obtain in large samples or may not be available in existing data sets [39]. Grimaud et al. (2014) used the Townsend material deprivation score, an index capturing French census-based measures of overcrowding, unemployment, and car and home ownership in a precise geographic unit, to study the impact of neighborhood-level social exposures on stroke mortality in Dijon, France [26]. The exposures comprising the Townsend score are primarily related to socioeconomic factors, with the distribution of these factors determined by the geographical and political powers driving the opportunities and resources available in a neighborhood. Aslanyan et al. (2003) used two other area-based indices, a categorical measure called a Womersly score, and a continuous measure known as a Murray score, to investigate the relationship between area-based deprivation and stroke subtype, severity, and outcomes in Scotland [19]. Both indices combine measures of housing, car ownership, occupation, and social status in each postal code, with an average population of 5000, to approximate the level of disadvantage in the area. They also incorporate measures for understanding how crime and access to health services vary across neighborhoods, thus making connections with geographic and political factors as well as health care factors. Brown et al. (2013) measured neighborhood socioeconomic status (NSES), an index developed for the Cardiovascular Health Study to determine if there are features of the neighborhood environment which contribute to mortality after stroke across four counties in the US [20]. The NSES was constructed from US Census data, which captured the median income, housing value, average resident education, and resident occupation, to rank the neighborhoods in a county into quartiles, from the least to most disadvantaged. Similarly, Nichols et al. (2018) used the socioeconomic index for areas to study the relationship between social disadvantage and aneurysmal subarachnoid hemorrhage incidence, while also incorporating an index capturing the rurality of where subjects lived with the accessibility/remoteness index of Australia [30].

The NIA framework includes individual socioeconomic factors within the environmental level of analysis. Rieske et al. (2004) investigated how the presence of vascular lesions and lesions indicative of Alzheimer’s disease varied across individual education groups, alluding to the role education may play in the processes contributing to cognitive decline [32]. Ojala-Oksala et al. (2012) used education history as a proxy for cognitive reserve and examined its relationship with stroke, post-stroke cognitive decline, and white matter lesions [31]. Lin et al. (1998) performed analyses of the prevalence of dementia subtypes in Taiwan, including vascular dementia diagnosed with the NINDS-AIREN criteria, stratified by level of education attained and further categorized participants as literate or illiterate [28]. These investigators also incorporated measures of occupation and rural or urban living area.

Race and ethnicity are considered fundamental factors for identifying priority populations within the NIA framework, rather than a factor within a particular level of analysis. We included studies using race and ethnicity as a primary exposure variable here because of the interactions between the exposure and socioeconomic factors examined by the investigators, placing it within the environmental level of analysis. An article from Waldstein et al. (2017) [36] aimed to study the interaction between race and socioeconomic status and their relationship to global brain outcomes assessed in white and African-American or Black participants. Sacco and colleagues (1998) [34] discuss how sampling from within a single community, as was done in the Northern Manhattan Stroke Study, from which they source their data for examining racial and ethnic differences in stroke incidence, helps their sample be more comparable in terms of the distribution of SES and access to health care services. Wolfe and colleagues (2002) [37] aimed for similar comparability in their analysis of first-ever strokes in two boroughs in South London, and also discussed the complex ways race and ethnicity can interact with SES.

#### 3.2.2. Sociocultural

Interestingly, only one article in our sample evaluated their exposures as determinants occurring at the sociocultural level. This level of analysis can include cultural factors, such as group values and norms; social factors, such as family stress and institutional racism; and psychological factors, such as stigma and bias [14]. Eshak et al. (2017) examined how changes in family home composition, as measured through the loss or gain of a family member in a particular living situation, could influence risk of stroke in Japan [25]. The authors discuss how losing a family member can impact psychological stress, financial stability, and social support, which all falls within the social factor category in the NIA framework. They also mention how the caregiving role is gendered in Japan, such that women may experience the negative effects of changes in family composition more than men, and that this may be reflected in differential rates of stroke between men and women.

#### 3.2.3. Behavioral and Biological

The NIA Health Disparities Research Framework suggests that the behavioral level of analysis is comprised of individual behaviors and processes, which can act as pathways between environmental and social exposures and health outcomes. For example, a health behavior can act as a coping mechanism for stressful life events. The biological level of analysis focuses on identifying biological processes acting as mechanisms for observed health disparities, such as chronic stress leading to the overload of hypothalamus-pituitary-adrenal axis, and ultimately leading to poor health outcomes. The eight remaining articles all identified the distribution of vascular risk factors seen in their study population across racial and ethnic groups. These risk factors typically used in the prediction of stroke events are age, systolic blood pressure, diabetes, cigarette smoking, atrial fibrillation, and left ventricular hypertrophy [40].

The article from Wright et al. (2017) captured how environmental, behavioral, and biological factors intersect in their study examining subclinical cerebrovascular disease as a risk factor for incident stroke mortality across Black, Hispanic, and white residents of a Manhattan, New York neighborhood [38]. This study utilized data from the Northern Manhattan Stroke Study, an ongoing cohort study that has enrolled over 4000 individuals living in the same community. While they were primarily interested in racial and ethnic differences in vascular risk factors, and later stroke, they also incorporated measures of education and insurance coverage, which may have also contributed to the disparities in stroke incidence found by the investigators. Labovitz et al. (2005) also worked with data from the Northern Manhattan Stroke Study to describe the incidence of intracerebral hemorrhage among the Black, white, and Hispanic study population and also examined the distribution of risk factors like hypertension and alcohol use [27]. In their analyses of racial and ethnic differences in microbleed prevalence, Copenhaver et al. (2008) similarly measured risk factors, comprised of both health behaviors and comorbid conditions [23]. The investigators acknowledged that one of the hospitals they sampled from tended to draw from a population which is considered medically underserved.

Romano et al. (2013) utilized prospective stroke registries in Miami, Florida, and Mexico City, Mexico to compare the nature and determinants of stroke across Hispanic populations [33]. These authors put greater emphasis on the behavioral risk factors in their analyses, which included both tobacco and drug use, but did not speculate as to why the distribution of these factors varied across ethnic groups. Sharma et al. (2012) examined the prevalence of different subtypes of stroke and related risk factors across a multi-ethnic Asian population in Singapore and suggested that the between group differences found could be related to additional, unmeasured health behaviors, such as dietary habits, as well as potential genetic differences [35].

The last three articles included in our review focus on a biological level of analysis, in which it is assumed that differences in comorbid conditions across racial or ethnic groups help explain differences in health outcomes. Bruno et al. (2000) examined clinical features of intracerebral hemorrhage between the Hispanic and non-Hispanic white populations in a single county in New Mexico and found that hypertension may be a greater risk factor for intracerebral hemorrhage in the Hispanic study population [21]. Chugunova and Nikolaeva (2013) were interested in the different features and risk factors of stroke among the indigenous and non-indigenous patients in a stroke registry in Yakutia, Russia [22]. Finally, Coull et al. (1990) investigated disparities in specific stroke type diagnoses among white and non-white patients in hospital-based stroke programs across the US [24]. They found that a significant proportion of those given a stroke but not otherwise specified diagnosis were non-white, even though their risk factor profiles were similar to those that did receive a particular stroke type diagnosis.

## 4. Discussion

Broadly, in this scoping review we sought to characterize studies on the relationship between measures of social disadvantage and confirmed cerebrovascular disease. While many studies have examined associations between social disadvantage and stroke incidence and hospitalizations, few have linked social exposures to the broader spectrum of cerebrovascular disease by confirming their outcomes using neuroimaging or autopsy. Nearly all of the included articles found a direct association between the social disadvantage measure and the cerebrovascular outcomes of interest, in that the part of the study populations exposed to social or economic disadvantage were more likely to be found to have confirmed cerebrovascular disease. Clearly, establishing the link between social disadvantage and cerebrovascular disease is an important step in defining the biological mechanisms that translate exposure into disease. This could help direct future health care practices and public health policies to reduce cerebrovascular disease and disparities.

We identified 20 studies which met our inclusion criteria. Our initial search yielded over 4000 results, the majority of which we excluded because they lacked confirmation of cerebrovascular disease by imaging or autopsy. A comprehensive assessment of cerebrovascular injury is critical for the identification of the causal mechanisms linking social disadvantage to disease. Thus, this area is rich with opportunities for future study. Sixteen out of the 20 studies in our final sample captured stroke as their primary outcome measure, either generally or focused on a particular subtype. While the prevalence of stroke in the general population warrants intense focus from the research community, there are additional measures of cerebrovascular disease that can be examined. Apart from large vessel events, small vessel disease, microbleeds, amyloid angiopathy, white matter hyperintensities, and perfusion abnormalities apart from stroke were underrepresented in the current review, suggesting that their relationship to social disadvantage remains less examined. All of the studies in our sample used CT or MRI to confirm cerebrovascular disease, either independently of or in addition to autopsy records or fluid biomarkers where appropriate.

Measures of social disadvantage differed across our included studies and covered varying levels of the NIA Health Disparities Research Framework [14]. Five articles used area-based indices capturing multiple factors of disadvantage, and three used education, all of which fall within the environmental level of analysis for this particular framework. Our results suggest that the literature could benefit from increased focus on the geographic and political, as well as the health care, factors in the environmental level of analysis, perhaps through collaboration with health geographers and policy analysts. We identified only one article that framed its research question through the sociocultural level of analysis, indicating that this could be an important area for future research as well. Involving experts, such as anthropologists and psychologists, or employing techniques like social network analysis, could help move the field forward in this area.

More than half of the articles used race or ethnicity as their primary exposure variable of cerebrovascular disease confirmed through imaging or autopsy. Three studies approached their research question through an environmental level of analysis, in which they discussed the complex interactions between race, socioeconomic, health care, and geographic and political factors [34,36,37]. It should be noted that while all three of these articles used race and ethnicity as an exposure, none of them made mention of racism, which shapes the distribution of the environmental factors explored in their studies [17].

The eight additional studies using race or ethnicity as their exposure did so through a biological and behavioral level of analysis. The authors focused on the way race or ethnicity impacted risk factors for cardiovascular disease such as smoking, hypertension, and diabetes and their associations with the outcomes of interest. However, we found several of the articles’ treatment of race and ethnicity to be inconsistent with the current consensus in the social scientific literature. Bruno and colleagues found that hypertension may be a greater risk factor for intracerebral hemorrhage in the Hispanic study population compared to non-Hispanic white participants but did not speculate as to why that could be [21]. Chugunova and Nikolaeva also found differences in stroke subtypes between the indigenous and non-indigenous populations in Yakutia, Russia, but did not discuss the implications or determinants of such differences [22]. The investigators in the article from Coull and colleagues found that a large proportion of those who did not receive a specific stroke diagnosis were non-white but did not attribute this disparity to race or other social factors [24]. Using race and ethnicity as a determinant for a biological outcome without situating it in a social context is troublingly similar to the racial essentialism described by Tsai et al. (2020) [41], in which race is used a proxy for genetic differences that are not present. Ross et al. (2020) [42] outlined a set of recommendations for researchers to consider when using race and ethnicity in their work, and which are essential for future investigations in this area. These include justifying the use of race and ethnicity variables in research, using these variables to describe social experiences rather than biology, and providing an interpretation of relevant findings beyond statistical significance.

The present study is not without its limitations. First, our initial search may have failed to identify all of the relevant articles in the literature. There may be articles published outside of our time frame, in an un-searched database, or in a language other than English that make significant contributions to the field. It is also possible that our choice of search terms may have unintentionally excluded articles meeting our criteria. Finally, our scoping review does not include an assessment of the quality of the included articles [15]. We primarily focused on how studies defined their exposures and outcomes, so the quality of the methodologies and conclusions of the articles reviewed here may vary.

## 5. Conclusions

This scoping review sought to assess the state of the literature examining the relationship between social disadvantage and confirmed cerebrovascular disease. It is clear that the confirmation of cerebrovascular disease through imaging and autopsy is not common in this area of research, but we assert that it is a necessary part of defining the underlying mechanism connecting social context to disease outcomes and may be a fruitful area for future research. Our findings suggest that there are elements of the NIA Health Disparities Research Framework that remain under-investigated, including geographic, political, and health care factors within the environmental level of analysis, and many factors within the sociocultural level of analysis. Additionally, work addressing how the dosage and timing of social disadvantage can impact the burden of cerebrovascular disease would further contribute to our understanding of the relationship between these constructs. Finally, we advocate careful consideration of the broad social context and clear justification when using race or ethnicity as a quantitative predictor. Continued contributions to this field are critical for improving our understanding of the mechanisms linking social factors and cerebrovascular disease, and to equitably improve disparities in brain health through advancements in health care practice and policy.

## Figures and Tables

**Figure 1 ijerph-18-07071-f001:**
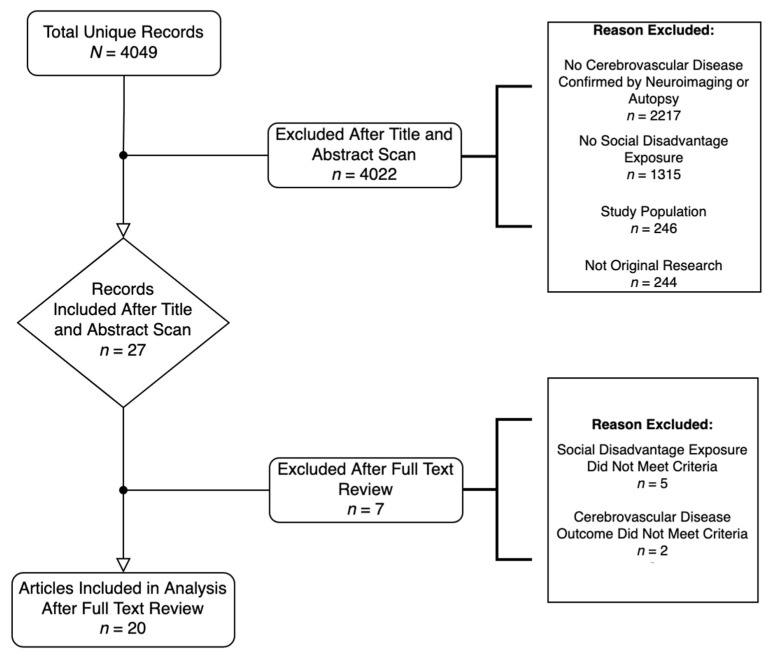
Article Selection Diagram.

**Table 1 ijerph-18-07071-t001:** Characteristics of Included Studies.

Study	Data Source	Sample	Measure of Social Disadvantage	Cerebrovascular Outcome	Main Findings
Aslanyan et al. (2003) [19]	Patient data from an acute stroke unit	Adult patients who were consecutively admitted to the Western Infirmary Acute Stroke Unit in Glasgow, Scotland between 1991 and 1998 (*N* = 2026)	Two area-based deprivation indices: Womersley and Murray scores	Total anterior circulation infarction, partial anterior circulation infarction, posterior circulation infarction, or lacunar infarction confirmed by CT or MRI	Higher area-based deprivation index scores were associated with stroke at younger age, higher systolic blood pressure, and more severe stroke.
Brown et al. (2013) [20]	Cardiovascular Health Study (CHS)	Random sample of non-institutionalized Medicare eligible patients over 65 from four counties across the United States between 1989 and 1992 (*N* = 3834)	Neighborhood socioeconomic status, race/ethnicity	Mortality after incident stroke confirmed by CT or MRI	In adjusted models, mortality hazard one year after stroke was significantly higher among residents of neighborhoods with the lowest socioeconomic status than those in the highest socioeconomic status neighborhood.
Bruno et al. (2000) [21]	Medical records	All cases of intracerebral hemorrhage among Hispanic and non-Hispanic white residents of a county in New Mexico, United States in 1993 (*N* = 84)	Race/ethnicity interacting with risk factors for stroke	Intracerebral hemorrhage confirmed by CT or autopsy	There were no statistically significant differences between the two ethnic groups in incidence of ICH or in prevalence of risk factors for stroke.
Chugunova and Nikolaeva (2013) [22]	Hospital stroke registry	Indigenous and non-indigenous Patients admitted to the stroke unit at the Department of Neurology of the Regional Vascular Centre in Yakutsk, Russia in 2011 (*N* = 1108)	Indigeneity to the Yakutsk, Russia region	Stroke types: ischemic stroke, intracerebral hemorrhages, subarachnoid hemorrhages, subarachnoid parenchymal hemorrhages, isolated intraventricular hemorrhages of non-traumatic etiology confirmed by CT or MRI	The share of a hemorrhagic stroke was higher in indigenous patients than in non-indigenous groups.
Copenhaver et al. (2008) [23]	Hospital stroke registry	Patients in two hospitals in the eastern United States diagnosed with intracerebral hemorrhage (*N* = 87)	Race/ethnicity interacting with behavioral risk factors for stroke	Microbleeds in primary intracerebral hemorrhage confirmed by MRI	Microbleeds were more prevalent in the Black patients, with 74% having at least one microbleed compared to 42% of white patients. The black population also tended to have a greater frequency of microbleeds in multiple territories compared to the white population. These differences were independent of risk factors like alcohol use.
Coull et al. (1990) [24]	Community hospital-based stroke programs	White and non-white patients hospitalized with stroke in community hospital-based stroke programs across three states in the United States (*N* = 4129)	Race/ethnicity interacting with behavioral risk factors for stroke	Infarction, hemorrhage, or stroke not otherwise specified confirmed by CT	Thirty percent of patients did not receive a specific stroke type diagnosis; these patients were often elderly, non-white, and had an altered level of consciousness at admission, but had a similar risk factor profile to that of patients who received a specific stroke type diagnosis.
Eshak et al. (2017) [25]	The Japan Public Health based Cohort (JPHC) study	Women and men aged 45–74 years in Japan (*N* = 140,420)	Household composition-gaining or losing a family member in a given living situation	Hemorrhagic and ischemic strokes and/or the presence of focal neurological deficits confirmed by CT, MRI, or autopsy	When compared with a stable family composition, losing at least one family member was associated with 11–15% increased risk of stroke in women and men.
Grimaud et al. (2014) [26]	Dijon StrokeRegistry	Incident strokes identified between 1998 and 2010 in Dijon, France (*N* = 1760)	Townsend neighborhood deprivation score	Ischemic, hemorrhagic, or undetermined stroke confirmed by CT or MRI	There was no association between deprivation and mortality while patients were in hospital care. After discharge, adjusted mortality gradually increased with deprivation score.
Labovitz et al. (2005) [27]	The Northern Manhattan Study	Incident intracerebral hemorrhage cases among adults in Northern Manhattan, United States between July 1993 and June 1997 (*N* = 155)	Race/ethnicity interacting with behavioral risk factors for stroke	Deep and lobar incident intracerebral hemorrhage confirmed by CT or autopsy	The incidence of intracerebral hemorrhage in Northern Manhattanwas greater for men than women and greater for Black and Caribbean Hispanic individuals than for whites. Most of the excess risk was for deep intracerebral hemorrhage in these groups. Smoking was more prevalent in the non-white participants, which could be a partial driver for differences in deep ICH.
Lin et al. (1998) [28]	Interviews and Mini-Mental Status Examination scores	Stratified random sample of a population in Southern Taiwan (*N* = 398)	Education, living area (urban/rural), and occupation	Vascular dementia diagnosed using NINDS AIREN criteria and confirmed with CT or MRI	Prevalence of dementia was significantly higher in people who were illiterate and higher in blue collar workers compared white collar workers. No significant difference between urban and rural populations.
McCormick and Chen (2016) [29]	South London Stroke Register (SLDR)	Patients with first ever strokes in South London, England between 1995 and 2011 (*N* = 782)	Carstairs Index of socioeconomic deprivation	Cerebral infarction, primary intracerebral hemorrhage, and subarachnoid hemorrhage confirmed by CT, MRI, or autopsy	A socioeconomic gradient was found, with the highest mortality after hemorrhagic stroke found in the most deprived quartile of the population.
Nichols et al. (2018) [30]	Statewide hospital administrative data	Non-traumatic aneurysmal subarachnoid hemorrhage cases in Tasmania, Australia (*N* = 237)	The Accessibility/Remoteness Index of Australia(ARIA), the Socioeconomic Index for Areas (SEIFA)	Aneurysmal subarachnoid hemorrhage confirmed by CT or MRI	A significant association between area-level socioeconomic disadvantage and aneurysmal subarachnoid hemorrhage incidence was observed, with the rate in disadvantaged geographical areas being 1.40 times higher than that in advantaged areas.
Ojala-Oksala et al. (2012) [31]	The Helsinki Stroke Aging Memory (SAM) cohort	A consecutive series of all patients with suspected stroke admitted to a hospital in Helsinki, Finland between 1993 and 1995 (*N* = 486)	Educational and marital history	Long-term survival post-acute stroke and neuropsychological deficits confirmed by MRI	Longer educational history was associated with lower frequency of executive dysfunction. Educational history was not associated with recurrent strokes, but it was associated with favorable post-stroke survival.
Riekse et al. (2004) [32]	University of Washington Alzheimer’s Disease Patient Registry (ADPR)	Subjects meeting clinical criteria for dementia with available clinical assessments in Seattle, WA, United States (*N* = 48)	Race/ethnicity interacting with education	Lesions indicative of Alzheimer’s disease alone and Alzheimer’s disease with vascular lesions confirmed by CT or autopsy	Alzheimer’s disease pathology with significant vascular lesions cases had higher baseline and final Mini-Mental State Examination scores than pure Alzheimer’s disease cases, but after adjusting for education, these differences were not statistically significant.
Romano et al. (2013) [33]	Prospective stroke registries	Hispanic patients admitted to one of the two study sites in Mexico City, Mexico and Miami, FL, United States with stroke (*N* = 928)	Race/ethnicity interacting with behavioral risk factors for stroke	Ischemic stroke, transient ischemic attack, primary intracerebral hemorrhage, and cerebral venous thrombosis confirmed by CT or MRI	Significant differences were found in the frequency in the different types of stroke and their severity between the patients in Mexico and Miami.
Sacco et al. (1998) [34]	The Northern Manhattan Study	Subjects diagnosed with first stroke between 1993 and 1996, residing in Northern Manhattan, United States (*N* = 662)	Race/ethnicity interacting with education	Cerebral infarction, intracerebral hemorrhage, or subarachnoid hemorrhage confirmed by CT	Black subjects had a 2.4-fold and Hispanic subjects a twofold increase in stroke incidence compared withwhite subjects which was not fully explained by differences in education.
Sharma et al. (2012) [35]	Hospital records	Patients with acute ischemic stroke admitted to a tertiary care hospital in Singapore between 2003 and 2004 (*N* = 481)	Race/ethnicity interacting with behavioral risk factors for stroke	Stroke subtypes, mortality functional independence, National Institutes of Health Stroke Score confirmed by CT or MRI	The prevalence of risk factors was similar in the three ethnic groups except for diabetes. Large-artery atherosclerotic infarctions were more prevalent in the Indian group (25.0%), whereas lacunar infarctions occurred more frequently in the Chinese group. No differences in in-hospital mortality and functional independence at discharge were seen among the three ethnic groups.
Waldstein et al. (2017) [36]	Healthy Aging in Neighborhoods of Diversity Across the Life Span, SCAN sub-study	White and African American adults aged 33 to 71 in Baltimore, MD, United States who participated in the parent study (*N* = 147)	Race/ethnicity interacting with a composite indicator of socioeconomic status, education, and income	White matter lesions, subclinical infarcts, and brain atrophy confirmed by MRI	Significant interactions of race and SES were observed for white matter lesion volume, total brain volume, and gray matter. African American participants with low SES exhibited significantly greater white matter lesion volumes than white participants with low SES.
Wolfe et al. (2002) [37]	South London Stroke Register (SLDR)	Patients with first-ever stroke in South London, England between 1995–1998 (*N* = 1254)	Race/ethnicity interacting with occupation	Cerebral infarction, primary intracerebral hemorrhage, and subarachnoid hemorrhage confirmed by CT, MRI, or autopsy	The Black patients in the sample were at increased risk for most stroke types, but there was no significant difference in post-stroke survival and occupation was not a significant factor in these differences.
Wright et al. (2017) [38]	The Northern Manhattan Study MRI sub-study	Stroke-free members of the initial Study cohort in Northern Manhattan, United States between 2003 and 2008 (*N* = 1287)	Race/ethnicity interacting with high school completion and insurance status	Incident stroke and mortality confirmed by MRI	Racial and ethnic variations in the effects of these subclinical brain findings were found independent of education and insurance status, suggesting differential risk of both stroke and mortality among these groups.

**Table 2 ijerph-18-07071-t002:** Included Studies Categorized by Level of Analysis of Social Disadvantage Measures.

Level of Analysis	Environmental	Sociocultural	Behavioral	Biological
Categories of Factors	Geographic and Political	Cultural	Coping	Physiological Indicators
	Socioeconomic	Social	Psychosocial Risk	Genetic Stability
	Health Care	Psychological	Health Behaviors	Cellular Function
Specific Factors	Area Deprivation	Household Composition	Smoking	Hypertension
	Education		Alcohol Use	Diabetes
	Insurance Coverage		Social Support	Systolic Blood Pressure
Articles of Interest	Aslanyan et al. (2003) [19]	Eshak et al. (2017) [25]	Copenhaver et al. (2008) [23]	Bruno et al. (2000) [21]
	Brown et al. (2013) [20]		Labovitz et al. (2005) [27]	Chugunova and Nikolaeva (2013) [22]
	Grimaud et al. (2014) [26]		Romano et al. (2013) [33]	Copenhaver et al. (2008) [23]
	Lin et al. (1998) [28]		Sharma et al. (2012) [35]	Coull et al. (1990) [24]
	McCormick and Chen (2016) [29]		Wright et al. (2017) [38]	Labovitz et al. (2005) [27]
	Nichols et al. (2018) [30]			Romano et al. (2013) [33]
	Ojala-Oksala et al. (2012) [31]			Sharma et al. (2012) [35]
	Riekse et al. (2004) [32]			Wright et al. (2017) [38]
	Sacco et al. (1998) [34]			
	Waldstein et al. (2017) [36]			
	Wolfe et al. (2002) [37]			

## Data Availability

Not applicable.

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
