# Peer review of "A Scoping Review of the Association of Social Disadvantage and Cerebrovascular Disease Confirmed by Neuroimaging and Neuropathology"

_ijerph, 2021, doi:10.3390/ijerph18137071_

Round 1

Reviewer 1 Report

Authors reviewed 4,049 titles and abstracts published in time period 1990-2020 which evaluated a measure of social disadvantage as an exposure variable and cerebrovascular disease 
confirmed through assessment of vascular neuropathology as an outcome measure. National Institute on Aging Health Disparities Research Framework was utilized to frame the findings. 20 records were deemed appropriate for full text review. Limitation of the study is that all reviewed studies have used different cerebrovascular disease outcome measures. Sociocultural, economic, educational and environmental background was investigated. A table with all studies and their main conclusions is provided. Novelty of this review is that it aims to link social exposures to the broader spectrum of 
cerebrovascular disease by confirming their outcomes using neuroimaging or autopsy. An additional table which summarize all the social factors which were found to be important throughout the literature and a proposal for future research would enrich this interesting review. Furthermore it would be of use to see if there is any data of higher incidence of surgically treated cerebrovascular disease dependant on the social status. 

Author Response

Response to Reviewer 1 Comments

POINT 1: An additional table which summarize all the social factors which were found to be important throughout the literature and a proposal for future research would enrich this interesting review.

RESPONSE 1: We thank Reviewer 1 for their comments guiding us towards a paper which is clearer for the readers and provides direction for future research. We agree that this paper could use additional clarity on the social factors identified in the literature. To assist with this, we added an additional row in Table 2 listing the specific factors of social disadvantage found in the included articles and where they fall within the NIA Health Disparities Research Framework.

We solidified our proposals for areas of future research in our conclusion section on page 12, paragraph 4, lines 377-87 with the following statement: “It is clear that the confirmation of cerebrovascular disease through imaging and autopsy is not common in this area of research, but we assert that it is a necessary part of defining the underlying mechanism connecting social context to disease outcomes and may be a fruitful area for future research. Our findings suggest that there are elements of the NIA Health Disparities Research Framework that remain under-investigated, including geographic, political, and health care factors within the environmental level of analysis, and any factor within the sociocultural level of analysis. Additionally, work addressing how the dosage and timing of social disadvantage can impact the burden of cerebrovascular disease would further contribute to our understanding of the relationship between these constructs.”

POINT 2: It would be of use to see if there is any data of higher incidence of surgically treated cerebrovascular disease dependent on the social status.

RESPONSE 2: While we agree that the surgical treatment of cerebrovascular disease is an area of concern with known disparities across different measures of social status, it is slightly beyond the scope of the current review. This would be an excellent area for future study, however.

Reviewer 2 Report

The authors reviews the association of social disadvantage, namely low-income, limited education, poor living condition, or limited social supports, with the vascular neuropathology. The review was done comprehensively and summarized in a clear way. Nevertheless, I'd like to raise some issues:

  1. The title implies that the authors will focus on the 'neuropathology'. Rather throughout the review, most included researches had clinical or neuroimaging outcome, but not actual 'pathology'. Therefore, the title word may need to be rephrased to avoid confusion.
  2. What are the main association the authors conclude? Are there positive association, or linear correlation between the social disadvantage and cerebrovascular diseases? Or merely the results are too heterogenous? Even in a narrative review, the readers may expect a more confirmative conclusion from the authors.
  3. Based on what order are the studies presented in the Table 1? It doesn't look like alphabetical nor chronological. Please clarify.
  4. It may be better to add on the Country name in the Table 1, probably in the 'data source' or 'sample' column.
  5. Since the authors stated that race or ethnicity alone does not constitute a 'social disadvantage', but rather their interaction with other factors does. Given that, it is better to add the 'interaction' variable in the Table, instead of only presented as 'race/ethnicity'.
  6. The 'cerebrovascular outcome' is still too heterogenous.  Some are purely clinical (stroke itself) while some are neuroimaging markers. It may be better to separate them into different searching strategies or to present in different tables. 
  7. In all tables, besides typing author name (year), it is suggest to add on the reference number. 

Author Response

Response to Reviewer 2 Comments

POINT 1: The title implies that the authors will focus on the 'neuropathology'. Rather throughout the review, most included research had clinical or neuroimaging outcome, but not actual 'pathology'. Therefore, the title word may need to be rephrased to avoid confusion.

RESPONSE 1: We thank Reviewer 2 for noticing this discrepancy between our initial title and the content of the paper. We have changed our title accordingly to “A Scoping Review of the Association of Social Disadvantage and Cerebrovascular Disease Confirmed by Neuroimaging and Neuropathology” to account for the fact that while some studies used assessment of neuropathology through autopsy to confirm the cerebrovascular disease of interest, most used some kind of neuroimaging technique, like CT or MRI scans, to confirm disease.

POINT 2: What are the main association the authors conclude? Are there positive association, or linear correlation between the social disadvantage and cerebrovascular diseases? Or merely the results are too heterogenous? Even in a narrative review, the readers may expect a more confirmative conclusion from the authors.

RESPONSE 2: We appreciate this insight into what readers of this special issue would like to see in a review such as ours. We added a statement clarifying the trend of the associations found in the included studies in our discussion section on page 11, paragraph 2, lines 304-7: “Nearly all of the included articles found a direct association between the social disadvantage measure and the cerebrovascular outcomes of interest, in that the part of the study populations exposed to social or economic disadvantage were more likely to be found to have confirmed cerebrovascular disease.”

POINT 3: Based on what order are the studies presented in the Table 1? It doesn't look like alphabetical nor chronological. Please clarify.

RESPONSE 3: Thank you for catching this oversight in Table 1’s organization. We have since sorted this table alphabetically by the last name of the first author of each paper.

POINT 4: It may be better to add on the Country name in the Table 1, probably in the 'data source' or 'sample' column.

RESPONSE 4: We agree that adding the country name to Table 1 would be useful for readers. We have added this information into the ‘Sample’ column of Table 1.

POINT 5: Since the authors stated that race or ethnicity alone does not constitute a 'social disadvantage', but rather their interaction with other factors does. Given that, it is better to add the 'interaction' variable in the Table, instead of only presented as 'race/ethnicity'.

RESPONSE 5: This is an excellent point. We have noted interaction among the race/ethnicity and other social disadvantage variables in the ‘measure of social disadvantage’ column of Table 1 to conserve space rather than adding a new column for this information.

POINT 6: The 'cerebrovascular outcome' is still too heterogenous.  Some are purely clinical (stroke itself) while some are neuroimaging markers. It may be better to separate them into different searching strategies or to present in different tables. 

RESPONSE 6: We agree that neuroimaging and pathology is an important focus for this work, and as such, we limited our search to only those studies which confirmed their cerebrovascular outcome with either imaging or autopsy, as discussed on page two, paragraph 3. Although some outcomes in the included studies are broader than others, each has been confirmed through one or both of these methods. We recognize that this was not made clear in Table 1 and have added how each outcome was confirmed in the ‘cerebrovascular outcome’ column of that table to help clarify this point.

POINT 7: In all tables, besides typing author name (year), it is suggested to add on the reference number.

RESPONSE 7: We thank the reviewer for pointing this out to us, and we concur that having the references in the tables would be beneficial for readers. We have added the relevant reference numbers to both Table 1 and 2 next to author name and year.

Round 2

Reviewer 2 Report

The authors answered my questions appropriately.